# Elevated CLOCK and BMAL1 Contribute to the Impairment of Aerobic Glycolysis from Astrocytes in Alzheimer’s Disease

**DOI:** 10.3390/ijms21217862

**Published:** 2020-10-23

**Authors:** Ik Dong Yoo, Min Woo Park, Hyeon Woo Cha, Sunmi Yoon, Napissara Boonpraman, Sun Shin Yi, Jong-Seok Moon

**Affiliations:** 1Department of Nuclear Medicine, Soonchunhyang University Hospital Cheonan, Cheonan 31151, Chungcheongnam-do, Korea; 92132@schmc.ac.kr; 2Department of Integrated Biomedical Science, Soonchunhyang Institute of Medi-bio Science (SIMS), Soonchunhyang University, Cheonan 31151, Chungcheongnam-do, Korea; pmw0269@sch.ac.kr (M.W.P.); ckgusdn88@naver.com (H.W.C.); 3Department of Biomedical Laboratory Science, College of Medical Sciences, Soonchunhyang University, Asan 31538, Chungcheongnam-do, Korea; sycshm0227@naver.com (S.Y.); beam_napissara@hotmail.com (N.B.)

**Keywords:** CLOCK, BMAL1, aerobic glycolysis, astrocytes, Alzheimer’s disease

## Abstract

Altered glucose metabolism has been implicated in the pathogenesis of Alzheimer’s disease (AD). Aerobic glycolysis from astrocytes is a critical metabolic pathway for brain energy metabolism. Disturbances of circadian rhythm have been associated with AD. While the role of circadian locomotor output cycles kaput (CLOCK) and brain muscle ARNT-like1 (BMAL1), the major components in the regulation of circadian rhythm, has been identified in the brain, the mechanism by which CLOCK and BMAL1 regulates the dysfunction of astrocytes in AD remains unclear. Here, we show that the protein levels of CLOCK and BMAL1 are significantly elevated in impaired astrocytes of cerebral cortex from patients with AD. We demonstrate that the over-expression of CLOCK and BMAL1 significantly suppresses aerobic glycolysis and lactate production by the reduction in hexokinase 1 (HK1) and lactate dehydrogenase A (LDHA) protein levels in human astrocytes. Moreover, the elevation of CLOCK and BMAL1 induces functional impairment by the suppression of glial fibrillary acidic protein (GFAP)-positive filaments in human astrocytes. Furthermore, the elevation of CLOCK and BMAL1 promotes cytotoxicity by the activation of caspase-3-dependent apoptosis in human astrocytes. These results suggest that the elevation of CLOCK and BMAL1 contributes to the impairment of astrocytes by inhibition of aerobic glycolysis in AD.

## 1. Introduction

Alzheimer’s disease (AD) is a progressive neurodegenerative disease characterized by impairment of memory and cognitive function [1]. Recent studies suggest that AD is a metabolic neurodegenerative disease [2,3,4]. The impairment in cerebral glucose metabolism is an invariant pathophysiological feature in AD [5,6]. The reduction in cerebral glucose metabolism is linked to AD progression and cognitive dysfunction [7,8,9]. Recent studies showed that the alteration of aerobic glycolysis is linked to the pathogenesis of AD [10,11].

Astrocytes are the most numerous subtypes of glial cell population in the central nervous system (CNS) [12,13]. The roles of astrocytes have been linked to brain development and function such as synapse formation and function, control of neurotransmitters release and uptake, production of trophic factors and control of neuronal survival [14,15,16,17,18]. In brain energy metabolism, the interaction of astrocytes with neuron and endothelial cells of the blood–brain barrier (BBB) plays a critical role in metabolic support to neurons through neurometabolic coupling including aerobic glycolysis, glutamate and Na–K-ATPase activation, and lactate release [19,20,21,22,23]. As a hub for neurometabolic and neurovascular coupling, astrocytes provide the function as gatekeepers of neuronal energy supply [24,25,26,27,28,29].

Glucose metabolism is an important metabolic pathway for growth and survival in many types of cells [30,31]. Previously, we have shown that the impairment of glycolysis induces apoptotic cell death [32]. Aerobic glycolysis from astrocytes is a critical metabolic pathway that processes the utilization of glucose to generate lactate as a metabolic intermediate which is used as a primary energy source for neurons [33]. In the aerobic glycolysis pathway, hexokinase 1 (HK1) is a first key enzyme for the initiation of glycolysis [34,35,36]. As the final step of anaerobic glycolysis, lactate is produced by conversion of pyruvate to lactate by lactate dehydrogenase (LDH) [37,38,39]. LDH is composed of four subunits. The two different subunits are LDHA (the M (muscle type) subunit of LDH) and LDHB (the H (heart type) subunit of LDH), which both retain the same active site and amino acids participating in the enzyme reaction [39,40]. Astrocytes express the protein of LDHA in the human brain [27,39,41,42].

Previous studies suggest that lactate production and transport between astrocytes and neurons is required for long-term memory formation [43,44]. In addition, lactate has also been identified as a novel metabolite that induces expression of genes associated with synaptic plasticity in the brain [45,46]. Disturbances of circadian rhythm have been associated with AD [47]. Recent evidence demonstrates that sleep–wake and circadian rhythm disturbances often occur early in the progress of AD [48,49]. Additionally, the disruption of circadian rhythm is linked to metabolic imbalance [50]. Circadian rhythm is mediated by the transcriptional–translational feedback loop of core clock genes (TTLs) including circadian locomotor output cycles kaput (CLOCK), brain muscle ARNT-like1 (BMAL1), period1 (PER1), PER2, cryptochrome1 (CRY1), and CRY2 [51,52,53,54]. CLOCK and BMAL1 proteins play in a positive feedback loop to regulate the expression of TTLs which are involved in diverse physiological process including major components of energy homeostasis like feeding behavior, locomotor activity, sleep–wake cycle and glucose utilization [55,56]. However, the mechanisms by which CLOCK/BMAL1 regulates aerobic glycolysis from astrocytes in AD remain unclear.

Here, we show that the protein levels of CLOCK and BMAL1 are significantly elevated in impaired astrocytes of cerebral cortex from patients with AD. We demonstrate that the overexpression of CLOCK and BMAL1 significantly suppresses aerobic glycolysis and lactate production by a reduction in HK1 and LDHA protein levels in human astrocytes. Moreover, the over-expression of CLOCK and BMAL1 induces the functional impairment by suppression of GFAP-positive filaments levels in human astrocytes. Furthermore, the elevation of CLOCK and BMAL1 promotes cytotoxicity by the activation of caspase-3-dependent apoptosis in human astrocytes. These results suggest that the elevated CLOCK and BMAL1 contribute to the impairment of aerobic glycolysis from astrocytes in AD.

## 2. Results

### 2.1. The Levels of CLOCK and BMAL1 Are Elevated in Impaired Astrocytes on Cortex Region from Patients with Alzheimer’s Disease

To investigate the role of CLOCK and BMAL1 in the dysfunction of astrocytes in patients with AD, we analyzed whether the protein levels of CLOCK and BMAL1 were elevated in the astrocytes surrounding the BBB of brains from patients with AD (Appendix A). We measured the protein levels of CLOCK and BMAL1 in GFAP-positive astrocytes surrounding the BBB on brain temporal cortex tissues from patients with AD or non-AD donor (normal) using immunofluorescence staining (Figure 1A,B). Immunofluorescence staining revealed that the intensity of CLOCK-positive staining in GFAP-positive astrocytes surrounding the BBB were increased in the molecular layer (ML) on the cortex region of patients with AD (AD) relative to non-AD donor (normal) (Figure 1A and Appendix A). Moreover, the number of GFAP-positive astrocytes surrounding the BBB was significantly less in ML on the cortex region of patients with AD (AD) relative to non-AD donor (Normal) (Figure 1A and Appendix A). Furthermore, the length of GFAP-positive filaments was shorter, and the shape of GFAP-positive filaments was more shrunk with AD (AD) compared to non-AD donor (Normal) (Figure 1A and Appendix A). Notably, the number of impaired astrocytes that have the positive subcellular co-localization between CLOCK and GFAP were significantly increased in patients with AD (AD) (Figure 1C). Consistent with CLOCK protein levels, the intensity of BMAL1-positive staining in GFAP-positive astrocytes surrounded BBB were increased in ML on cortex region of patients with AD (AD) relative to the non-AD donor (normal) (Figure 1B and Appendix A). The number of impaired astrocytes that have the positive subcellular co-localization between BMAL1 and GFAP were significantly increased in patients with AD (AD) (Figure 1D). The levels of CLOCK and BMAL1 were generally higher in every individual patient with AD. These results suggest that the CLOCK and BMAL1 protein levels in impaired astrocytes surrounding the BBB were elevated in patients with AD.

### 2.2. The Elevation of CLOCK and BMAL1 Suppresses Aerobic Glycolysis and Lactate Production by Reduction in HK1 and LDHA Protein Levels in Human Astrocytes

We investigated the underlying molecular mechanism by which the elevation of CLOCK and BMAL1 induces the impairment of astrocytes during AD. We examined whether the elevation of CLOCK and BMAL1 could suppress aerobic glycolysis in human astrocytes. We analyzed the changes of aerobic glycolysis activity by over-expression of CLOCK and BMAL1 in human astrocytes. We measured the extracellular acidification rate (ECAR) as the parameter of aerobic glycolysis activity by the quantification of lactate production. The glycolytic activity was measured by the sequential addition of glucose, which is the substrate of glycolysis, oligomycin, a selective inhibitor for mitochondrial respiration, and 2-deoxyglucose (2-DG), a specific inhibitor of glycolysis. We first examined the change of ECAR levels by CLOCK over-expression in human astrocytes (Figure 2A,B). The levels of ECAR in response to glucose were significantly suppressed by CLOCK over-expression relative to control (Figure 2A,B). We next investigated the molecular target in the regulation of aerobic glycolysis pathway by CLOCK over-expression in human astrocytes. We analyzed the levels of hexokinase 1 (HK1), which is a first key enzyme for the initiation of glycolysis, and LDHA, which is a critical enzyme to produce lactate, in human astrocytes (Figure 2C,F). Consistent with the reduction in glycolytic activity, HK1 and LDHA protein levels were significantly suppressed by CLOCK over-expression compared to control (Figure 2C). Similarly, BMAL1 over-expression significantly suppressed the levels of ECAR in response to glucose relative to control (Figure 2D,E). Moreover, the protein levels of HK1 and LDHA were suppressed by BMAL1 over-expression compared to control (Figure 2F). These results suggest that the elevation of CLOCK and BMAL1 suppresses aerobic glycolysis and lactate production by reduction in HK1 and LDHA protein levels in human astrocytes.

### 2.3. The Elevation of CLOCK and BMAL1 Contributes to the Functional Impairment by Reduction in GFAP-Positive Filaments in Human Astrocytes

Next, we investigated the role of CLOCK and BMAL1 in the functional impairment of astrocytes in AD. We examined whether the elevation of CLOCK and BMAL1 could induce the functional impairment by reduction in GFAP-positive filaments, which plays a role in astrocyte–neuron interactions as well as cell–cell communication, in human astrocytes. We first analyzed the changes of GFAP-positive filaments and cellular morphology by CLOCK and BMAL1 over-expression using immunofluorescence staining (Figure 3A,B and Appendix A). Notably, immunofluorescence staining revealed that the intensity of GFAP-positive filaments was significantly reduced by CLOCK over-expression compared to control (Figure 3A). Moreover, the length of GFAP-positive filaments in cytosol was shorter, and the shape of GFAP-positive filaments was shrunk by CLOCK over-expression relative to control (Figure 3A). Similarly, BMAL1 over-expression suppressed the intensity of GFAP-positive filaments (Figure 3B). BMAL1 over-expression induced the shortness and shrinkage of GFAP-positive filaments relative to control (Figure 3B). Consistent with immunofluorescence staining, the protein levels of GFAP were significantly reduced by CLOCK and BMAL1 over-expression compared to control (Figure 3C,D). These results suggest that the elevation of CLOCK and BMAL1 induces the functional impairment by reduction in GFAP-positive filaments in human astrocytes.

### 2.4. The Elevation of CLOCK and BMAL1 Induces Cytotoxicity in Human Astrocytes

We next investigated the role of CLOCK and BMAL1 in cytotoxicity of astrocytes in AD. Since the elevation of CLOCK and BMAL1 resulted in the impairment of aerobic glycolysis and GFAP-positive filaments in human astrocytes, we examined whether the elevation of CLOCK and BMAL1 could induce the cell death. We analyzed the morphological changes of cytotoxicity by over-expression of CLOCK and BMAL1 using 3D analyzer in human astrocytes. Notably, CLOCK over-expression increased the morphological features of cytotoxicity including shrinkage and blebbing relative to the control (Figure 4A). Moreover, the compartment of cytosolic organelles such as mitochondria was disrupted by over-expression of CLOCK relative to the control (Figure 4A). Furthermore, the number of morphological dead cells by CLOCK over-expression was significantly increased compared to the control (Figure 4C). Consistent with the morphological analysis, the over-expression of CLOCK significantly increased the levels of cytotoxicity relative to the control (Figure 4C). Similarly, BMAL1 over-expression increased the morphological features of cytotoxicity compared to the control (Figure 4B). The number of morphological dead cells and the levels of cytotoxicity were significantly increased by BMAL1 over-expression relative to the control (Figure 4D). These results suggest that the elevation of CLOCK and BMAL1 induces the cytotoxicity in human astrocytes.

### 2.5. The Elevation of CLOCK and BMAL1 Induces Caspase-3-Dependent Apoptosis in Human Astrocytes

Next, we investigated the molecular mechanism by which the elevation of CLOCK and BMAL1 promotes cell death of astrocytes in AD. We examined whether the over-expression of CLOCK and BMAL1 could induce caspase-3-dependent apoptosis in human astrocytes. We measured the levels of cleaved caspase-3 as an active form of capase-3 in the apoptosis pathway. Notably, the over-expression of CLOCK significantly increased the protein levels of cleaved caspase-3 compared to control (Figure 5A). Consistently, the over-expression of BMAL1 significantly increased the levels of cleaved caspase-3 relative to control (Figure 5B). These results suggest that the elevation of CLOCK and BMAL1 induces caspase-3-dependent apoptosis in human astrocytes. In summary, our results demonstrate that elevated CLOCK and BMAL1 induce the dysfunction and cytotoxicity of astrocytes via the impairment of aerobic glycolysis in Alzheimer’s disease (Figure 5C).

## 3. Discussion

Here, we demonstrate that elevation of CLOCK and BMAL1 contributes to the impairment of astrocytes in AD. We suggest that the elevation of CLOCK and BMAL1 induces the impairment of aerobic glycolysis and GFAP-positive filaments in human astrocytes. Our findings provide a molecular mechanism by which the elevation of CLOCK and BMAL1 is critical for the metabolic and functional dysfunction of astrocytes in the pathogenesis of AD.

Disturbances of circadian rhythm have been associated with AD [57,58,59]. Recent study suggests that disturbances of circadian rhythm often occur at an early stage in the progress of AD as an important clinical feature [60,61]. However, the underlying molecular mechanism by which disturbances of circadian rhythm affect the metabolic and functional impairment of astrocytes in AD is still not fully understood. Our results suggest that the elevation of CLOCK/BMAL1 in astrocytes represents the disruption of interactions of astrocytes with neurons and BBB by disturbances of circadian rhythm during AD. Similarly, BMAL1 gene expression in suprachiasmatic nucleus (SCN) of hypothalamus, which is the principal circadian pacemaker, was significantly increased in 18-month-old AD mice compared to 18-month-old non-AD mice in response to 12 h exposure to darkness in light and dark cycle test [62]. Consistent with previous study, our results indicate that the abnormal expression of BMAL1 was observed in brain of patients with AD. Since the previous study showed the alteration of circadian rhythm gene expression in AD mice [62], further study for the changes in CLOCK/BMAL1 protein expression in brain tissues of AD mice would need to be studied.

As the most important energy regulator, astrocytes are well known in the brain [20]. Aerobic glycolysis from astrocytes is a critical metabolic pathway for lactate production and supply of lactate to neurons using glucose in blood vessels [41,63,64]. Although the role of aerobic glycolysis and lactate production in astrocytes has been demonstrated, the molecular targets that trigger the impairment of aerobic glycolysis and lactate production in astrocytes during AD are not well understood. Our results suggest that CLOCK/BMAL1 could be a critical molecule in the regulation of aerobic glycolysis and lactate production in astrocytes during AD. Similarly, the analysis by fluorodeoxyglucose (FDG)-positron emission tomography (PET) (FDG-PET) imaging revealed that the glucose utilization was reduced in the brains of patients with AD [65,66]. These previous studies suggest that FDG-PET imaging might be an important diagnostic tool that increases diagnostic accuracy and confidence for AD [65,66]. Consistent with previous study, our findings indicate that the dysfunction of aerobic glycolysis from astrocytes might be linked to the lower glucose uptake in FDG-PET imaging of patients with AD. In addition to FDG-PET imaging, the sleep test, which measures both sleep quality and sleep quantity, would need to be a diagnostic tool of AD. We anticipate the sleep test to be more appealing for use as a diagnostic tool in human AD in the future.

As well as the metabolic function, astrocytes have important roles, such as physical support, biochemical support and repair and scarring of injury for the BBB and neurons [19,20,67,68,69]. In these functions of astrocytes, GFAP is proposed to play a role in astrocyte–neuron interactions and cell–cell communication [70]. In addition, GFAP has been shown to be important in repair after injury in brain [70]. Our results show that the reduction in GFAP-positive filaments is linked to the morphological dysfunction of astrocytes, which have the high levels of CLOCK/BMAL1, in AD patients and human astrocytes. Our findings indicate that the elevation of CLOCK/BMAL1 could be a potential mechanism of the disruption of GFAP-mediated interaction of astrocytes with BBB and neurons in AD. Although we showed the reduction in GFAP levels by CLOCK/BMAL1 over-expression in human astrocytes, there is a limitation to support whether the elevation of CLOCK/BMAL1 can directly affect the impairment of GFAP-mediated interaction of astrocytes in patients with AD. Further study for the role of CLOCK/BMAL1 elevation on the GFAP-mediated interaction of astrocytes in patients with AD would need to be studied.

In the function of astrocytes, the role of reactive astrocytes in the contribution to plaque formation and maturation or to amyloid-beta (Aβ) clearance and plaque growth restriction is still unclear [71,72]. Our results indicate that the impaired astrocytes, which have the short and shrunk GFAP-positive filaments, have the high levels of CLOCK/BMAL1 expression in the brains of patients with AD. Although we found the morphological impairment of astrocyte in patients with AD, it is unclear whether these impaired astrocytes play a role as reactive astrocytes or dysfunctional astrocytes in AD. Further study for the regulation and role of CLOCK/BMAL1 expression in reactive astrocytes in vitro would need to be studied.

The astrocytic apoptosis is associated with both senile plaques and neurofibrillary tangles (NFTs) in AD [73]. Consistent with this previous study, our results suggest that the elevation of CLOCK/BMAL1 contributes to the activation of caspase-3-dependent apoptosis in astrocytes. Since the inhibition of glycolysis induces apoptotic cell death [74,75], our results indicate that the impairment of aerobic glycolysis by CLOCK/BMAL1 over-expression could be a potential upstream mechanism of caspase-3-dependent apoptosis in astrocytes.

In our study, we demonstrate that the elevation of CLOCK/BMAL1 could be an important molecular mechanism in the dysfunction of astrocytes during the pathogenesis of AD.

## 4. Materials and Methods

### 4.1. Human Subject Study

Our human subject study was conducted in accordance with the Helsinki Declaration. The protocol was approved by the Institutional Review Board of Soonchunhyang University Hospital Cheonan (SCHCA 2020-03-030-001). A total of 12 paraffin embedded brain tissues including frontal cortex, occipital cortex, temporal cortex and parietal cortex from 3 donors with Alzheimer’s disease were obtained from the Netherlands Brain Bank. A total of 4 paraffin embedded adult normal brain tissues including frontal cortex (NBP2-77761), occipital cortex (NBP2-77766), temporal cortex (NBP2-77774) and parietal cortex (NBP2-77769) were obtained from Novus Biologicals (Minneapolis, MN, USA).

### 4.2. Reagents and Antibodies

The following antibodies were used: polyclonal rabbit anti-CLOCK (ab3517, Abcam, (Cambridge, UK)), polyclonal rabbit anti-BMAL1 antibody (NB100-2288, Novus Biologicals), monoclonal mouse anti-GFAP (#3670, Cell signaling technology (Danvers, MA, USA)), polyclonal rabbit anti-Caspase-3 (#9662, Cell signaling Technology (Danvers, MA, USA)), and monoclonal mouse anti-β-actin (A5316, Sigma-Aldrich (St Louis, MO, USA)). Fluoroshield™ with DAPI (F6057, Sigma-Aldrich (St Louis, MO, USA)) was used for nuclear staining and mounting. Sections were mounted onto gelatin-coated slides with Canada Balsam (Wako, Tokyo, Japan) following dehydration.

### 4.3. Human Astrocytes

Human astrocytes were obtained from N7805100 (Thermo Fisher Scientific (Waltham, MA, USA)) and #1800 (ScienCell Research Laboratories (Carlsbad, CA, USA)). Human astrocytes are normal human cells derived from human brain tissue. Human astrocytes were cultured in Gibco™ Astrocyte Medium containing N-2 Supplement, Dulbecco’s modified Eagle medium (DMEM), 10% (vol/vol) One Shot™ Fetal Bovine Serum (FBS), 100 units/mL penicillin and 100 mg/mL streptomycin (A1261301, Thermo Fisher Scientific (Waltham, MA, USA)). For overexpression of human CLOCK or BMAL1, human astrocytes were seeded and transduced with pCMV6-AC-GFP constructs against human CLOCK (NM_004898) (RG221408, Origene (Rockville, MD, USA)), pCMV6-AC-GFP constructs against human BMAL1 (ARNTL) (NM_001178) (RG207870, Origene (Rockville, MD, USA)) or pCMV6-AC-GFP vector (PS100010, Origene).

### 4.4. Immunofluorescence Analysis

For immunofluorescence analysis, brain tissues were sectioned from paraffin embedded tissue blocks at a thickness of 4 μm. Sections were permeabilized in 0.5% Triton-X (T8787, Sigma-Aldrich (St Louis, MO, USA)), blocked in CAS-Block™ Histochemical Reagent (008120, Thermo Fisher Scientific (Waltham, MA, USA)), and then stained with the following antibodies: polyclonal rabbit anti-CLOCK (1:100) (ab3517, Abcam (Cambridge, U.K.)), polyclonal rabbit anti-BMAL1 antibody (1:100) (NB100-2288, Novus Biologicals (Minneapolis, MN, USA)), monoclonal mouse anti-GFAP (1:100) (#3670, Cell signaling technology (Danvers, MA, USA)). The secondary antibody was goat anti-rabbit IgG (H+L) Alexa Fluor 488 (A11008, Thermo Fisher Scientific (Waltham, MA, USA)), and goat anti-mouse IgG H&L Texas Red (ab6787, Abcam (Cambridge, U.K.)) for 2 h at 25 °C, respectively. Fluoroshield™ with DAPI (F6057, Sigma-Aldrich (St Louis, MO, USA)) was used for nuclear staining. Stained brain sections were analyzed by THUNDER Imager Tissue (Leica Microsystems Ltd. (Wetzlar, Germany)). Stained brain sections were quantified by LAS X image-processing software (Leica Microsystems Ltd. (Wetzlar, Germany)) and ImageJ software v1.52a (Bethesda, MD, USA). To ensure objectivity, all measurements were performed under blind conditions by two observers per experiment under identical conditions. Cells were plated and treated on autoclaved glass coverslips placed in 6-well cell culture plates. Cells were fixed in 4% PFA, permeabilized in 0.5% Triton-X (T8787, Sigma-Aldrich (St Louis, MO, USA)), blocked in CAS-Block™ Histochemical Reagent (008120, Thermo Fisher Scientific (Waltham, MA, USA)), and then stained as detailed above.

### 4.5. Glycolysis Activity Assay

For the glycolytic function assay, human astrocytes (5 × 10^4^ cells/well) were plated on XF96 cell culture microplates (101085-004, Agilent Technologies, Inc. (Santa Clara, CA, USA)). The ECAR, which is a parameter of glycolytic flux and activity, was measured by a Seahorse XF96e bioanalyzer using the XF Glycolysis Stress Test Kit (102194-100, Agilent Technologies, Inc. (Santa Clara, CA, USA)) according to the manufacturer’s instructions. The ECAR levels were monitored and measured in cells that were treated with glucose (10 mM), oligomycin (2 μM) and 2-deoxyglucose (2DG) (10 mM).

### 4.6. 3D Images

Human astrocytes (2 × 10^4^ cells) were seeded in FluoroDishTM (FD35-100, World Precision Instruments (Sarasota, FL, USA)). Cell were transduced with pCMV6-AC-GFP constructs against human CLOCK (NM_004898) (RG221408, Origene (Rockville, MD, USA)), pCMV6-AC-GFP constructs against human BMAL1 (ARNTL) (NM_001178) (RG207870, Origene (Rockville, MD, USA)) or pCMV6-AC-GFP vector (PS100010, Origene). Cells were incubated for 24 h or 48 h. Three-dimensional images were analyzed by 3D Cell Explorer (NANOLIVE (Ecublens, Switzerland)). The images were representative images from a total of 100 cells in ten individual images per group.

### 4.7. Immunoblot Analysis

Cells were harvested and lysed in NP40 Cell Lysis Buffer (FNN0021, Thermo Fisher Scientific (Waltham, MA, USA)). Lysates were centrifuged at 15,300× *g* for 10 min at 4 °C, and the supernatants were obtained. The protein concentrations of the supernatants were determined by using the Bradford assay (500-0006, Bio-Rad Laboratories (Hercules, CA, USA)). Proteins were electrophoresed on NuPAGE 4–12% Bis-Tris gels (Thermo Fisher Scientific (Waltham, MA, USA)) and transferred to Protran nitrocellulose membranes (10600001, GE Healthcare Life science (Pittsburgh, PA, USA)). Membranes were blocked in 5% (*w*/*v*) bovine serum albumin (BSA) (9048-46-8, Santa Cruz Biotechnology (Dallas, TX, USA)) in TBS-T (TBS (170-6435, Bio-Rad Laboratories (Hercules, CA, USA)) and 1% (*v*/*v*) Tween-20 (170-6531, Bio-Rad Laboratories) for 30 min at 25 °C. Membranes were incubated with primary antibody (1:1000) as detailed above diluted in 1% (*w*/*v*) BSA in TBS-T for 16 h at 4 °C and then with the horseradish peroxidase (HRP)-conjugated secondary antibody (goat anti–rabbit IgG–HRP (SC-2004) (1:2500) and goat anti-mouse IgG–HRP (SC-2005) (1:2500) from Santa Cruz Biotechnology (Dallas, TX, USA)) diluted in TBS-T for 0.5 h at room temperature. The immunoreactive bands were detected by the SuperSignal West Pico Chemiluminescent Substrate (34078, Thermo Scientific (Waltham, MA, USA)).

### 4.8. Cell Cytotoxicity Assay

Cell cytotoxicity was measured from culture medium of human astrocytes by LDH-Cytotoxicity Colorimetric Assay Kit II (#K313-500, BioVision (Milpitas, CA, USA)) following the manufacturer’s instructions. Human astrocytes (2 × 10^5^ cells in 6-well cell culture plate) were seeded and transduced with pCMV6-AC-GFP constructs against human CLOCK (NM_004898) (RG221408, Origene (Rockville, MD, USA)), pCMV6-AC-GFP constructs against human BMAL1 (ARNTL) (NM_001178) (RG207870, Origene (Rockville, MD, USA)) or pCMV6-AC-GFP vector (PS100010, Origene). Cells were incubated for 24 h.

### 4.9. Statistical Analysis

All data are represented as mean ± standard deviation (SD) or standard error of the mean (SEM). All statistical tests were analyzed using a two-tailed Student’s *t*-test for comparison of two groups, and analysis of variance (ANOVA) (with post hoc comparisons using Dunnett’s test) using a statistical software package (GraphPad Prism version 8.0, GraphPad Software Inc. (San Diego, CA, USA)) for comparison of multiple groups. *p* values (*, *p* < 0.05, **, *p* < 0.01, ***, *p* < 0.001) were considered statistically significant.

## 5. Conclusions

Consequently, we found the role of CLOCK/BMAL1 elevation in the impairment of astrocytes in AD. Particularly, we showed that the elevation of CLOCK/BMAL1 contributes to the reduction in aerobic glycolysis and GFAP-positive filaments in human astrocytes. Our results suggest that the elevation of CLOCK/BMAL1 could be a critical molecular mechanism in the dysfunction of astrocytes in the pathogenesis of AD.

## Figures and Tables

**Figure 1 ijms-21-07862-f001:**
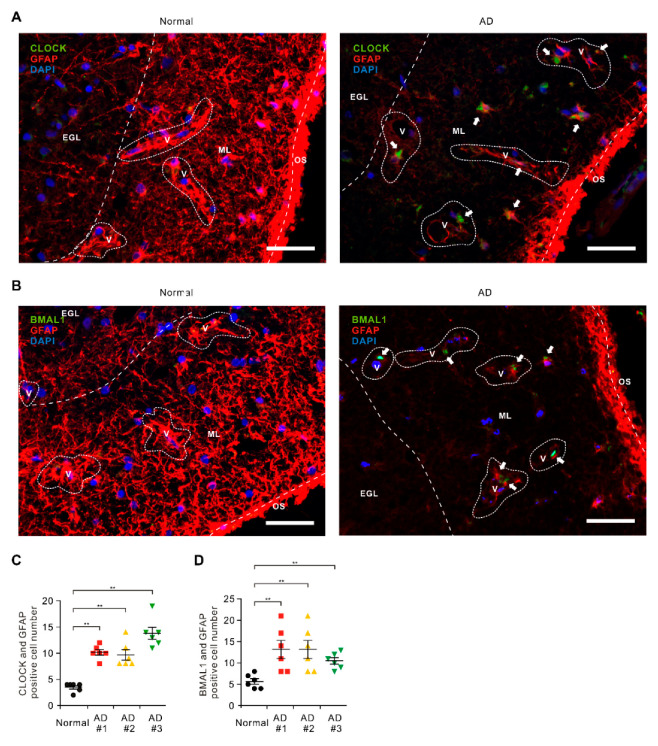
The levels of circadian locomotor output cycles kaput (CLOCK) and brain muscle ARNT-like1 (BMAL1) are elevated in impaired astrocytes in cerebral cortex region from patients with Alzheimer’s disease (AD). (**A**) Representative immunofluorescence images of CLOCK expression in cerebral cortex region from patients with AD (AD) or non-AD (normal) showing CLOCK (green) in astrocytes expressing the astrocytes marker GFAP (red) around blood vessels (V) (*n* = 3 per group, *n* = 6 images per individual subject). DAPI-stained nuclei are shown in blue. Outer surface (OS); molecular layer (ML); external granular layer (EGL). Scale bars, 20 μm. (**B**) Representative immunofluorescence images of BMAL1 expression in cerebral cortex region from patients with AD (AD) or non-AD (normal) showing BMAL1 (green) in astrocytes expressing the astrocytes’ marker GFAP (red) around blood vessels (V) (*n* = 3 per group, *n* = 6 images per individual subject). DAPI-stained nuclei are shown in blue. Outer surface (OS); molecular layer (ML); external granular layer (EGL). Scale bars, 20 μm. DAPI-stained nuclei are shown in blue. (**C**) Quantification of CLOCK positive astrocytes from immunofluorescence images in cerebral cortex region from patients with AD (AD) or non-AD (normal) (*n* = 3 per group, *n* = 6 images per individual subject). Data are mean ± standard deviation (SD). **, *p* < 0.01 by Student’s two-tailed *t*-test. (**D**) Quantification of BMAL1 positive astrocytes from immunofluorescence images in cerebral cortex region from patients with AD (AD) or non-AD (normal) (*n* = 3 per group, *n* = 6 images per individual subject). Data are mean ± standard deviation (SD). Symbols, which are expressed by white dotted line, indicate the outline of BBB. **, *p* < 0.01 by Student’s two-tailed *t*-test.

**Figure 2 ijms-21-07862-f002:**
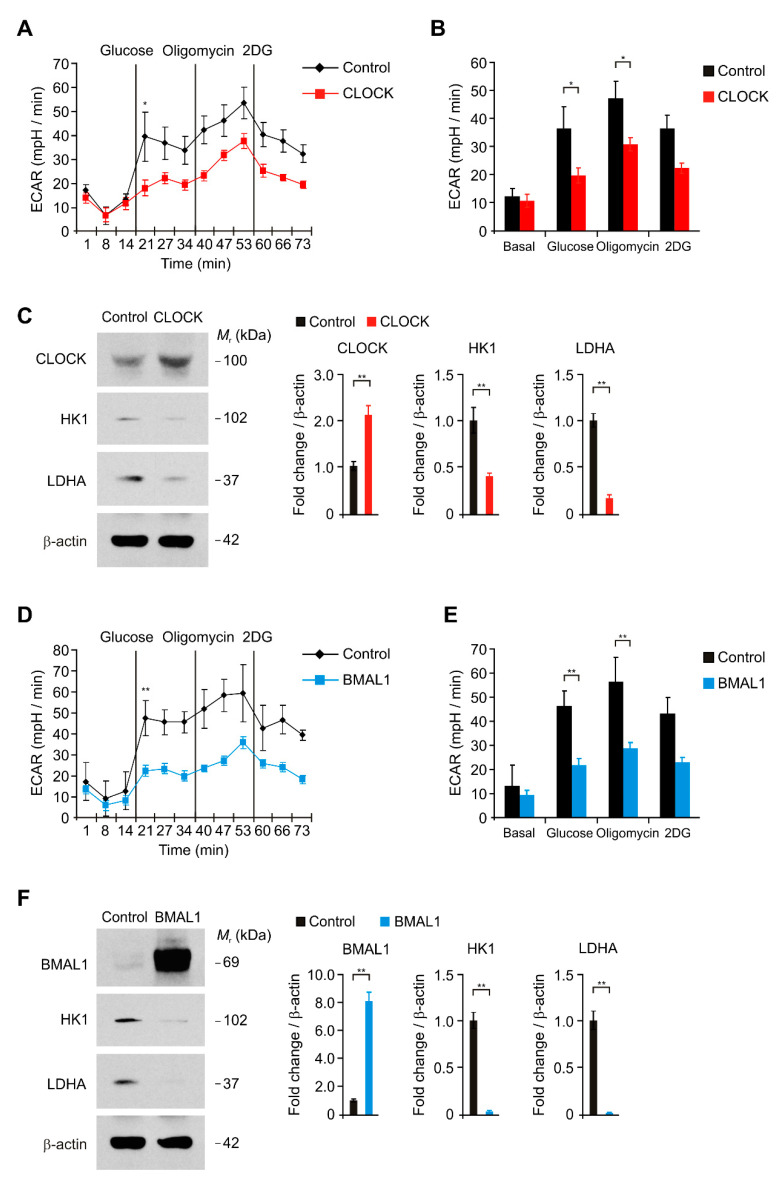
The elevation of CLOCK and BMAL1 suppresses aerobic glycolysis and lactate production by reduction in hexokinase 1 (HK1) and lactate dehydrogenase A (LDHA) protein levels in human astrocytes. (**A**) The levels of extracellular acidification rate (ECAR) for glycolysis of glucose and (**B**) quantification of ECAR levels in control and CLOCK-overexpressing (CLOCK) human astrocytes. Data are representative of three independent experiments. Data are mean ± SEM. * *p* < 0.05 using the two-tailed Student’s *t*-test. (**C**) Representative immunoblot analysis for CLOCK, HK1, LDHA (left) and quantification for CLOCK, HK1, LDHA protein levels (right) in control and CLOCK-overexpressing (CLOCK) human astrocytes. For immunoblots, β-actin was used as loading control. Data are representative of three independent experiments. Data are mean ± SEM. ** *p* < 0.01 using the two-tailed Student’s *t*-test. (**D**) The levels of extracellular acidification rate (ECAR) for glycolysis of glucose and (**E**) quantification of ECAR levels in control and BMAL1-overexpressing (BMAL1) human astrocytes. Data are representative of three independent experiments. Data are mean ± SEM. * *p* < 0.05 using the two-tailed Student’s *t*-test. (**F**) Representative immunoblot analysis for BMAL1, HK1, LDHA (left) and quantification for BMAL1, HK1, LDHA protein levels (right) in control and BMAL1-overexpressing (BMAL1) human astrocytes. For immunoblots, β-actin was used as loading control. Data are representative of three independent experiments. Data are mean ± SEM. ** *p* < 0.01 using the two-tailed Student’s *t*-test.

**Figure 3 ijms-21-07862-f003:**
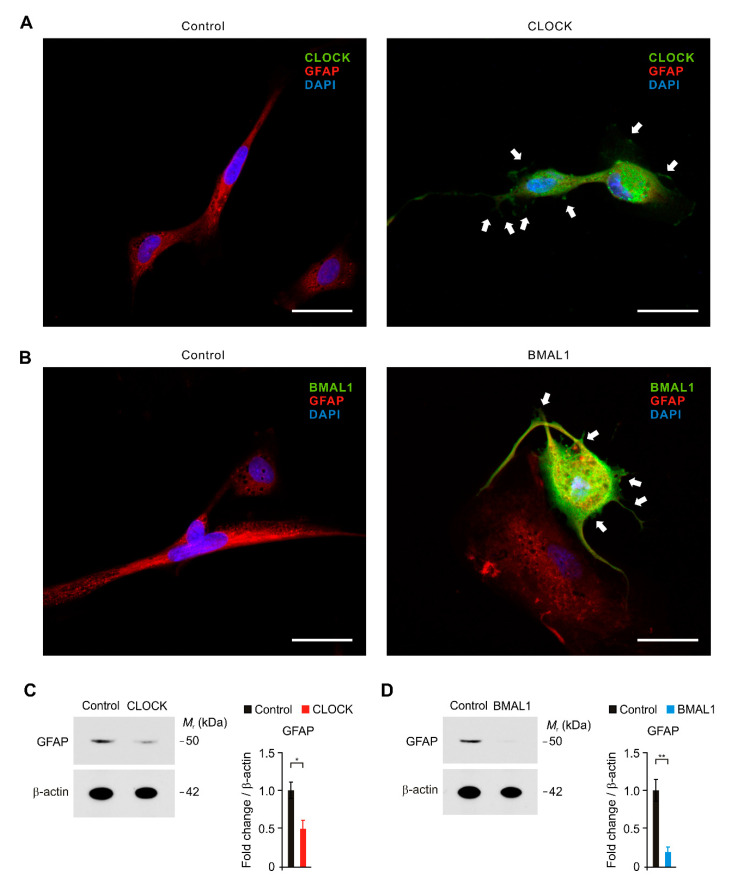
The elevation of CLOCK and BMAL1 contributes to the functional impairment by reduction in GFAP-positive filaments in human astrocytes. (**A**) Representative immunofluorescence images of CLOCK expression in control and CLOCK-overexpressing (CLOCK) human astrocytes showing CLOCK (green) and GFAP (red) (*n* = 10 per group). DAPI-stained nuclei are shown in blue. The shortness and shrinkage of GFAP-positive filaments were indicated (white arrows). Scale bars, 20 μm. (**B**) Representative immunofluorescence images of BMAL1 expression in control and BMAL1-overexpressing (BMAL1) human astrocytes showing BMAL1 (green) and GFAP (red) (*n* = 10 per group). DAPI-stained nuclei are shown in blue. The shortness and shrinkage of GFAP-positive filaments were indicated (white arrows). Scale bars, 20 μm. (**C**) Representative immunoblot analysis for GFAP (left) and quantification for GFAP protein levels (right) in control and CLOCK-overexpressing (CLOCK) human astrocytes. For immunoblots, β-actin was used as loading control. Data are representative of three independent experiments. Data are mean ± SEM. * *p* < 0.05 using the two-tailed Student’s *t*-test. (**D**) Representative immunoblot analysis for GFAP (left) and quantification for GFAP protein levels (right) in control and BMAL1-overexpressing (BMAL1) human astrocytes. For immunoblots, β-actin was used as loading control. Data are representative of three independent experiments. Data are mean ± SEM. * *p* < 0.05, ** *p* < 0.01 using the two-tailed Student’s *t*-test.

**Figure 4 ijms-21-07862-f004:**
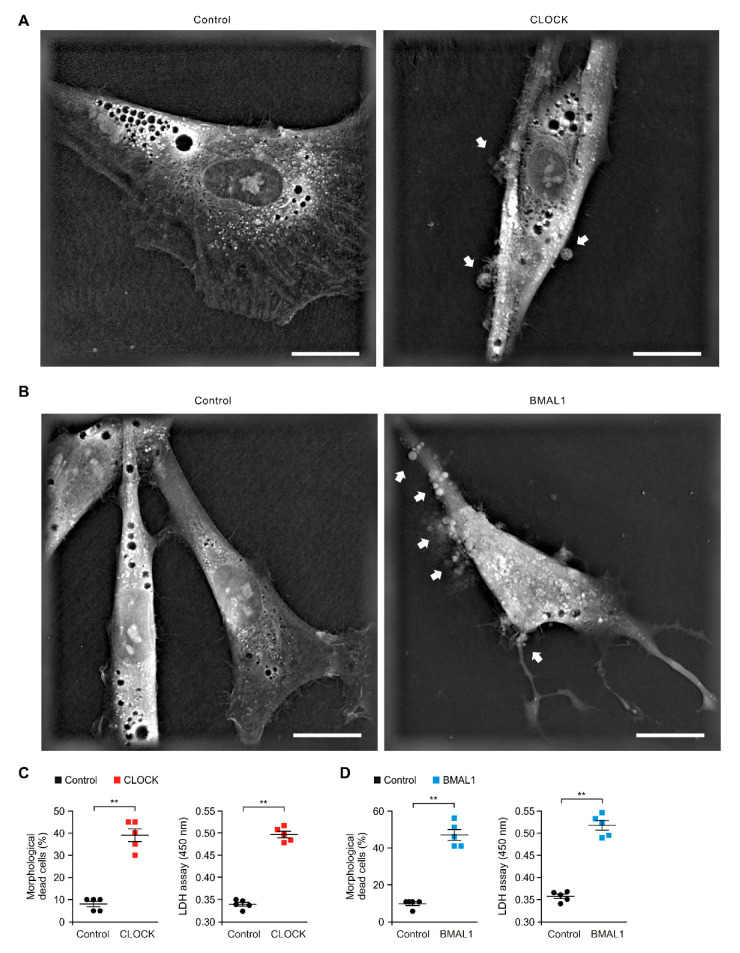
The elevation of CLOCK and BMAL1 induces cytotoxicity in human astrocytes. (**A**) Representative 3D images of control and CLOCK-overexpressing (CLOCK) human astrocytes (*n* = 10 images per group). The morphological features of cytotoxicity were indicated (white arrows). Scale bars, 20 μm. (**B**) Representative 3D images of control and BMAL1-overexpressing (BMAL1) human astrocytes (*n* = 10 images per group). The morphological features of cytotoxicity were indicated (white arrows). Scale bars, 20 μm. (**C**) Quantification of the morphological dead cells (left) in control and CLOCK-overexpressing (CLOCK) human astrocytes (the percent of morphological dead cells in total 100 cells in 10 individual images per group). Cytotoxicity assay (right) in control and CLOCK-overexpressing (CLOCK) human astrocytes using lactate dehydrogenase (LDH) levels. Data are representative of three independent experiments, and each was performed in triplicate. Data are mean ± SEM. ** *p* < 0.01 using the two-tailed Student’s *t*-test. (**D**) Quantification of the morphological dead cells (left) in control and BMAL1-overexpressing (BMAL1) human astrocytes (the percent of morphological dead cells in total 100 cells in 10 individual images per group). Cytotoxicity assay (right) in control and BMAL1-overexpressing (BMAL1) human astrocytes using lactate dehydrogenase (LDH) levels. Data are representative of three independent experiments, and each was performed in triplicate. Data are mean ± SEM. ** *p* < 0.01 using the two-tailed Student’s *t*-test.

**Figure 5 ijms-21-07862-f005:**
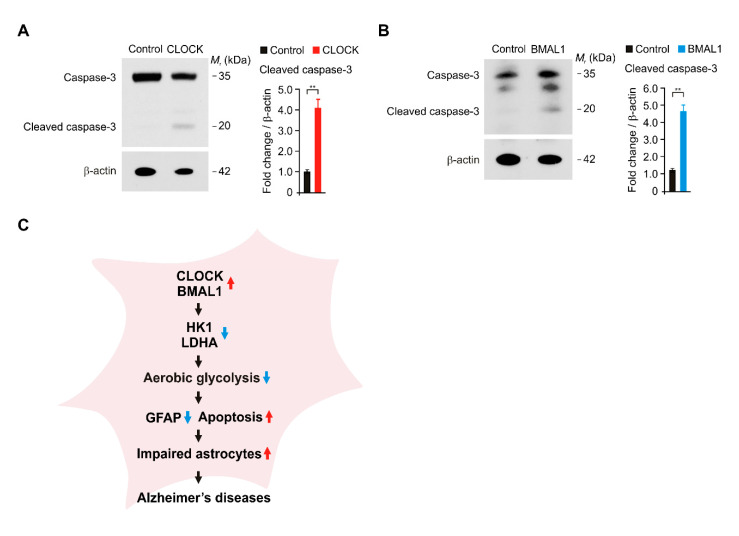
The elevation of CLOCK and BMAL1 induces caspase-3-dependent apoptosis in human astrocytes. (**A**) Representative immunoblot analysis for cleaved caspase-3 and total caspase-3 (left) and quantification for cleaved caspase-3 protein levels (right) in control and CLOCK-overexpressing (CLOCK) human astrocytes. For immunoblots, β-actin was used as loading control. Data are representative of three independent experiments. Data are mean ± SEM. ** *p* < 0.01 using the two-tailed Student’s *t*-test. (**B**) Representative immunoblot analysis for cleaved caspase-3 and total caspase-3 (left) and quantification for cleaved caspase-3 protein levels (right) in control and BMAL1-overexpressing (BMAL1) human astrocytes. For immunoblots, β-actin was used as loading control. Data are representative of three independent experiments. Data are mean ± SEM. ** *p* < 0.01 using the two-tailed Student’s *t*-test. (**C**) A schematic diagram to summarize our new findings. Red arrow means the increase. Blue arrow means the decrease in a diagram.

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
