# Peer review of "Elevated CLOCK and BMAL1 Contribute to the Impairment of Aerobic Glycolysis from Astrocytes in Alzheimer’s Disease"

_ijms, 2020, doi:10.3390/ijms21217862_

Round 1
Reviewer 1 Report
The author investigated the mechanism by which CLOCK and BMAL1 regulates the dysfunction of astrocytes in Alzheimer's Disease and found that the elevated CLOCK and BMAL1 contributes to the impairment of aerobic glycolysis from astrocytes. I congratulate the authors with the scientific soundness of the experiments and results.
Major comments:
1. The results, although very promising, remain in the basic research field. Could the authors add to the discussion, whether similar experiments have been performed in animals, and whether they support the authors results?
2. What is the potential clinical relevance of the authors discovery? Could the authors elaborate on the potential application of the results?
Minor comment: Please have the English language checked for spelling and grammar.
Author Response
Response to IJMS Reviewer 1 Comments
The author investigated the mechanism by which CLOCK and BMAL1 regulates the dysfunction of astrocytes in Alzheimer's Disease and found that the elevated CLOCK and BMAL1 contributes to the impairment of aerobic glycolysis from astrocytes. I congratulate the authors with the scientific soundness of the experiments and results.
Major comments:
Reviewer’s Comment 1
- The results, although very promising, remain in the basic research field. Could the authors add to the discussion, whether similar experiments have been performed in animals, and whether they support the authors results?
Response 1
As reviewer’s comment, we tried to discuss the regulation of CLOCK and BMAL1 in mouse model of AD in the previous study in Discussion section. We provided the more description for the changes of CLOCK and BMAL1 in AD mice.
The following text has been added to Page 11. Line 270:
Page 11. Line 270 “Similarly, BMAL1 gene expression in suprachiasmatic nucleus (SCN) of hypothalamus which is the principal circadian pacemaker was significantly increased in 18-month-old AD mice compared to 18-month-old non-AD mice in response to 12 h exposure to darkness in light and dark cycle test [REF 1]. Consistent with previous study, our results indicate that the abnormal expression of BMAL1 was observed in human AD brain. Since the previous study showed the alteration of circadian rhythm gene expression in AD mice [REF 1], further study for the changes of CLOCK/BMAL1 protein expression in brain tissues of AD mice would need to be studied.”
Reviewer’s Comment 2
- What is the potential clinical relevance of the authors discovery? Could the authors elaborate on the potential application of the results?
Response 2
As reviewer’s comment, we provided additional description for the potential clinical relevance of our findings in Discussion section. We tried to discuss the analysis of FDG-PET imaging in AD patients related to our findings. Also, we provided the description for the potential application related to our findings.
The following text has been added to Page 11. Line 283:
Page 11. Line 283 “Similarly, the analysis by fluorodeoxyglucose (FDG)-positron emission tomography (PET) (FDG-PET) imaging revealed that the glucose utilization was reduced in brain of patients with AD [REF2,3]. These previous studies suggest that FDG-PET imaging might be an important diagnostic tool that increases diagnostic accuracy and confidence for AD [REF2,3]. Consistent with previous study, our findings indicate that the dysfunction of aerobic glycolysis from astrocytes might be linked to the lower glucose uptake in FDG-PET imaging of patients with AD. In addition to FDG-PET imaging, the sleep test, which measures both sleep quality and sleep quantity, would need to be a diagnostic tool of AD. We anticipate the sleep test to be more appealing for use in human AD in the future.”
Reviewer’s Comment 3
Minor comment: Please have the English language checked for spelling and grammar.
Response 3
As reviewer’s comment, we will have the English editing by the journal.
Reviewer 2 Report
Alzheimer’s disease (AD) is a neurodegenerative disease and often accompanied with disruption in the sleep-wake cycle and circadian rhythms. Astrocytes play an important role in brain health and neurovegetative diseases including AD, astrocyte activation was observed in brains from patients with AD. In this manuscript, the authors investigated the role of two core circadian clock proteins, CLOCK and BMAL1, in the astrocyte dysfunction in AD. They show that elevated levels of CLOCK and BMAL1 lead to aerobic glycolysis impairment and that HK1 and LDHA protein levels were decreased, which in turn, affect the lactate production. This manuscript is well structured and written in a good standard English language. I just have few questions for the authors.
- It has been shown that ablation of BMAL1 in astrocytes promotes neuronal death in vitro, so I was wondering whether the elevated levels of BMAL1 and CLOCK occur in response to astrocyte activation and could play a protective role? Could the authors comment on this.
- Are there any significant variations in BMAL1 and CLOCK levels between the three AD cases?
- Have the authors measured the enzymatic activity for HK1 and LDHA? To see if there was any inhibition?
- Typo error, in page 2, line 47, in for, should be in the
- In references list ref 45, the numbering was duplicated
- The authors used AD to refer to Alzheimer’s disease, they do not need to repeat it everywhere in the manuscript, and the same case for the other abbreviations.
Author Response
Response to IJMS Reviewer 2 Comments
Alzheimer’s disease (AD) is a neurodegenerative disease and often accompanied with disruption in the sleep-wake cycle and circadian rhythms. Astrocytes play an important role in brain health and neurovegetative diseases including AD, astrocyte activation was observed in brains from patients with AD. In this manuscript, the authors investigated the role of two core circadian clock proteins, CLOCK and BMAL1, in the astrocyte dysfunction in AD. They show that elevated levels of CLOCK and BMAL1 lead to aerobic glycolysis impairment and that HK1 and LDHA protein levels were decreased, which in turn, affect the lactate production. This manuscript is well structured and written in a good standard English language. I just have few questions for the authors.
Reviewer’s Comment 1
It has been shown that ablation of BMAL1 in astrocytes promotes neuronal death in vitro, so I was wondering whether the elevated levels of BMAL1 and CLOCK occur in response to astrocyte activation and could play a protective role? Could the authors comment on this.
Response 1
As reviewer’s comment, we discussed the role of BMAL1 and CLOCK elevation in terms of astrocyte activation related to our findings in Discussion section.
The following text has been added to Page 11. Line 305:
Page 11. Line 305 “In the function of astrocytes, the role of reactive astrocytes in the contribution to plaque formation and maturation or to amyloid-beta (Aβ) clearance and plaque growth restriction is still unclear [REF4,5]. Our results indicate that the impaired astrocytes, which has the short and shrinked GFAP-positive filaments, have the high levels of CLOCK/BMAL1 expression in brain of patients with AD. Although we found the morphological impairment of astrocyte in patients with AD, it is unclear whether these impaired astrocytes play a role as reactive astrocytes or dysfunctional astrocytes in AD. Further study for the regulation and role of CLOCK/BMAL1 expression in reactive astrocytes in vitro would need to be studied.”
Reviewer’s Comment 2
Are there any significant variations in BMAL1 and CLOCK levels between the three AD cases?
Response 2
As reviewer’s comment, we provided additional description for the expression of BMAL1 and CLOCK in patients with AD in Results section.
The following text has been added to Page 3. Line 105:
Page 3. Line 105 “The levels of CLOCK and BMAL1 were generally higher in every individual patient with AD.”
Reviewer’s Comment 3
Have the authors measured the enzymatic activity for HK1 and LDHA? To see if there was any inhibition?
Response 3
We agree with reviewer’s comment for the measurement of HK1 and LDHA activity. Currently, the real-time metabolic activity analysis in live cells using Seahorse XF96e bioanalyzer is a representative assay for HK1 and LDHA activity by glycolysis stress test with glucose, oligomycin and 2-deoxyglucose. In our study, we measured the glycolytic activity in human astrocytes as live cells using Seahorse XF96e bioanalyzer. We analyzed the extracellular acidification rate (ECAR) as the parameter of glycolysis activity by the quantification of lactate production levels. As the measurement of ECAR induction after glucose supply compared to basal condition, we can evaluate the HK1 activity. As the measurement of total amount for ECAR during glycolysis stress test, we can evaluate the LDHA activity. As reviewer’s comment, we would like to try the individual measurement for enzymatic activity of HK1 and LDHA using protein lysates from human astrocytes in further study.
Reviewer’s Comment 4
Typo error, in page 2, line 47, in for, should be in the
Response 4
As reviewer’s comment, we revised Typo error in Page 2. Line 47.
The following text has been added to Page 2. Line 48:
Page 2. Line 48 “in the metabolic support to neurons through neurometabolic coupling including aerobic glycolysis”
Reviewer’s Comment 5
In references list ref 45, the numbering was duplicated
Response 5
As reviewer’s comment, we revised the numbering for ref 45 in Page 17. Line 554.
Reviewer’s Comment 6
The authors used AD to refer to Alzheimer’s disease, they do not need to repeat it everywhere in the manuscript, and the same case for the other abbreviations.
Response 6
As reviewer’s comment, we checked abbreviations in our manuscript.